# Secondary Metabolites and Antioxidant Activity of the Solid-State Fermentation in Apple (*Pirus malus* L.) and Agave Mezcalero (*Agave angustifolia* H.) Bagasse

**DOI:** 10.3390/jof6030137

**Published:** 2020-08-18

**Authors:** Diego Ibarra-Cantún, María Elena Ramos-Cassellis, Marco Antonio Marín-Castro, Rosalía del Carmen Castelán-Vega

**Affiliations:** 1Posgrado en Ciencias Ambientales, Benemérita Universidad Autónoma de Puebla, Col. Jardines de San Manuel, Edificio IC6, 72570 Puebla, Mexico; diego.ibarrac@alumno.buap.mx; 2Facultad de Ingeniería Química, Benemérita Universidad Autónoma de Puebla, Av. San Claudio y 18 Sur, 72570 Puebla, Mexico; 3Departamento de Investigación en Ciencias Agrícolas, Benemérita Universidad Autónoma de Puebla, 14 Sur 6301 Edificio IC1, 72570 Puebla, Mexico; marco.marin@correo.buap.mx (M.A.M.-C.); rosalia.castelan@correo.buap.mx (R.d.C.C.-V.)

**Keywords:** agave mezcalero bagasse, apple bagasse, solid-state fermentation, secondary metabolites, *Pleurotus ostreatus*

## Abstract

Solid-state fermentation (SSF) is used in enzyme and antibiotic production, bioethanol and biodiesel as an alternative energy source, biosurfactants with environmental goals, and the production of organic acids and bioactive compounds. The present project determined the quantity of secondary metabolites and the antioxidant activity of the extracts obtained by the solid-state fermentation of apple and agave mezcalero bagasse over 28 days, inoculated with the *Pleurotus ostreatus* strain. The extraction was carried out with three solvents: acetone and water (80:20 *v/v*), 100% methanol and 100% water. The results showed a higher presence of phenolic compounds, flavonoids, total triterpenes and antioxidant activity in the apple bagasse from the SSF on day 21 in the extract of acetone and water (80:20 *v/v*), 100% methanol and aqueous; while the agave bagasse showed a significant presence of phenolic compounds and flavonoids only in the aqueous extract. In conclusion, the presence of secondary metabolites exhibiting antioxidant activities from the solid-state fermentation in the residues of the cider and mezcal industry is an alternative use for wasted raw material, plus, it reduces the pollution generated from the agroindustrial residues.

## 1. Introduction

The residues generated by the agroindustry are, in most cases, not processed, causing environmental pollution and adverse effects on health [1]. In Mexico, this situation is mainly reflected in the production of the cider industry in Puebla and the production of mezcal in the state of Oaxaca. The former reported an annual production of 150 thousand bottles [2], where the processed fruit residues fluctuate between 15% and 30% of said production, depending on the state of maturation and the used technology [3]. The latter generated 122,696 tons of bagasse in 2012, which is usually dumped in rivers or streams, endangering the region’s environment [4].

Most of these wastes degrade slowly due to their solid and organic nature, which corresponds to their lignocellulosic biomass, rich in cellulose, hemicellulose and lignin. In addition, they are not subjected to an adequate process of exploitation, giving as a result an utterly deficient disposal of the waste [5,6]. Therefore, the aim of the present investigation was to determine the content of the total phenolic compounds, total flavonoids and total triterpenes, and the antioxidant activity, by 2,2-diphenyl-1-picrylhydrazyl (DPPH•) and 2,2′-azino-bis-3-ethylbenzothiazoline-6-sulfonic acid radical cation (ABTS●+) test methods resulting from the solid-state fermentation of apple bagasse and agave mezcalero bagasse substrates inoculated with the *Pleurotus ostreatus* strain, in order to optimize the utilization of residues of the cider and mezcalero industry in the attaining of secondary metabolites.

*Pleurotus ostreatus* synthesizes bioactive compounds, such as: polysaccharides (alpha and beta glucan) and polyphenols [7,8]. The macromycetes, such as *Pleurotus* and *Agaricus*, contain tetracyclic triterpenoids, with the sterols being the most abundant ones. At the same time, in *Pleurotus mutilis* and *Clitopilus passeckerianus* (called *Pleurotus passeckerianus* in the past), pleuromutilin and diterpene were identified as having antibiotic action against microplasmatic infections in animals [9,10].

Singhania et al. [11] and Thomas et al. [12] describe solid-state fermentation (SSF) as a process that takes place in a solid matrix that is within an inert support or substrate, or near free water, with just enough to promote the growth and metabolic activity of microorganisms. The SSF reduces the production costs due to the lower energy consumption and production area (smaller reactors), proving that it is an economic, interesting and environmentally friendly process for the production of secondary metabolites from agroindustrial and forest residues [13]. The SSF has generated great interest in recent years due to its variety of uses, such as bioremediation, biofuels (biodiesel, bioethanol, biobutanol, and biohydrogen among others) and the production or extraction of bioactive compounds [14].

One of the main applications of the solid fermentation of agroindustrial wastes is the extraction of secondary metabolites, also called phytochemicals, which include carotenes, terpenes, sterols, tocopherols and polyphenols. These can be used in the treatment and prevention of various diseases [15], since they have antitumor, anti-inflammatory, antithrombotic, antimicrobial, immune system modulating, and hypoglycemic properties; also, they reduce the concentration of lipids in the blood and display antioxidant activities [16,17]. Waste from agribusiness, such as the peel of fruits and vegetables, has been employed recently as a potential source of production of bioactive compounds that are commonly known as phenolic compounds [18]. Therefore, the secondary metabolites obtained from the solid fermentation of residues can be used as natural bioactive triggers for the formulation of functional foods or serve as additives in food products to extend their shelf life [19].

## 2. Materials and Methods

### 2.1. Obtaining Plant Material

The apple bagasse was collected from Huejotzingo’s cider industry, a town located in the state of Puebla, Mexico, and the agave mezcalero bagasse was collected from San Pablo Villa de Mitla’s mezcal industry, a town located in the state of Oaxaca, Mexico.

These bagasses were transported in airtight containers at 20 °C in less than 4 h. They were subsequently bleached at 70 °C for 5 min and dried in a forced air dryer (HTP 72 Lumistell Celaya, Gto. México) for 8 h at 60 °C until the moisture content was less than 10%. Once dried, the bagasse was crushed in a mill to obtain a particle of 0.841 mm to be used during solid fermentation.

Physicochemical tests were performed on the crushed bagasse in order to discover its attributes and adjust its physicochemical properties to make them suitable for the growth of the *Pleurotus* fungus. The bagasse reconditioning was carried out according to Staments and Chilton [20] where it is mentioned that *Pleurotus ostreatus* culture must have the following conditions: a humidity percentage of 80–90%, temperature from 23 to 26 °C, and pH from 6.5 to 7.5. The temperature and relative humidity were controlled during the process. The physicochemical tests performed were: total soluble solids (°Brix) [21], pH [21], moisture percentage [21], water activity (A*w*) [21], and titratable acidity percentage [21]; the meq value of malic acid were 0.067, the meq value of acetic acid were 0.060 and direct reducing sugars were also used [21].

### 2.2. Solid-State Fermentation

This process was accomplished with the contribution of the Mycology Laboratory of the Departamento de Investigación de Ciencias Agrícolas (Agricultural Sciences Research Department) of the Benemérita Universidad Autónoma de Puebla (Benemeritus Autonomous University of Puebla), which provided the *Pleurotus ostreatus* strains.

The solid-state fermentation was carried out in 9.7 × 2.6 cm glass tubes with a Bakelite cork, and each contained 10 g of each residue adjusted to a humidity percentage of 80% and a pH of 6.5, which was then sterilized in an autoclave ((CV300 AESA Tecámac, Edo. México, México)) at 120 °C and 15 psi for 45 min.

Then, a 1 cm diameter mycelium-agar circle of the *Pleurotus ostreatus* strain, corresponding to 3 mg of dry weight biomass, was incorporated into each tube and incubated in an oven (E51 Riossa Monterrey, N.L., México) at 25 °C. Samples were taken at 0, 7, 14, 21 and 28 days.

### 2.3. Solid Waste Extraction

After the fermentation days, 12 mL of each solvent was added to each glass tube for extraction. The used solvents were acetone and water (80:20 *v/v*), 100% methanol and distilled water. The samples were placed in a shaker (KJ-201BD Orbital Shaker Westtune Hangzhou, Zhejiang, China) at 80 rpm for 1 h, filtered and placed in a centrifuge (Z200A Hermle Wehingen, Germany) at 47 g, and the ensuing supernatant was stored in Eppendorf tubes at −20 °C for further analysis.

### 2.4. Quantification of Total Phenolic Compounds (TPC)

The analysis was performed using the method described by Singleton and Rossi [22], with some modifications. A total of 250 µL of 50% Folin–Ciocalteu reagent was added to 50 µL of the extract; then it was stirred and allowed to stand in the dark for 8 min. Subsequently, 1.25 mL of 5% CaCO_3_ (*w/v*) was added and allowed to stand again for 30 min in the dark at room temperature. An absorbance at 725 nm was read afterwards on a UV-Vis spectrophotometer (7305 Jenway Staffordshire, UK). A calibration curve for gallic acid (Sigma Aldrich, CAS 149-91-7, Steinheim, Germany) was used in the range of 0 to 0.3 g/L. The results were expressed in mg gallic acid equivalent per g substrate on dry weight (mg GAE/g substrate dw). These analyses were performed in triplicate.

### 2.5. Total Flavonoid Quantification (TF)

The methodology proposed by Chang et al. [23] was used, with some modifications. A total of 500 µL of the extract, 1.5 mL of 80% ethanol (*v/v*), 100 µL of 10% hexahydrate aluminum chloride solution, 100 µL of 1 M potassium acetate, and 2.8 mL of distilled water were placed, stirred and incubated for 30 min at room temperature.

Subsequently, an absorbance at 415 nm was read on a UV-Vis spectrophotometer (7305 Jenway Staffordshire, UK). A quercetin calibration curve (Sigma Aldrich, CAS 117-39-5, Germany) was used at a concentration of 0 to 0.1 g/L. The results were expressed in mg quercetin equivalents per g substrate on dry weight (mg QE/g substrate dw). These analyzes were performed in triplicate.

### 2.6. Total Triterpenes Quantification (TT)

The vanillin-acetic acid method was used, with some modifications [24]. A total of 120 μL of each extract, 100 μL of vanillin (Sigma Aldrich, CAS 121-33-5, Germany) 5% (*w/v*), and 400 μL of perchloric acid in 0.1 N glacial acetic acid were placed together in a test tube at 60 °C for 15 min.

Then, the mixture was cooled down to room temperature and 2.5 mL of glacial acetic acid were added. The absorbance was measured at 550 nm in a UV-Vis spectrophotometer (7305 Jenway Staffordshire, UK). The results were expressed as mg ursolic acid per g substrate on dry weight (mg UA/g substrate dw). A calibration curve for uric acid (Sigma Aldrich, CAS 77-52-1, Germany) was used with a concentration from 0 to 120 g/L. These analyses were performed in triplicate.

### 2.7. Antioxidant Activity by 1,1-Diphenyl-2-Picril Hydracil (DPPH•)

The DPPH• (2,2-Diphenyl-1-picrilhydrazyl) radical in methanol solution was developed by Brand-Williams et al. [25], and was used with some alterations. A total of 990 μL of DPPH (Sigma Aldrich, CAS 1898-66-4, Germany) 0.188 mM was added to 10 μL of the extract. The mixture was then homogenized and kept in the dark for 30 min at room temperature.

The absorbance at 517 nm was measured in both the control (methanol (A0)) and the test samples (A1) on a UV-Vis spectrophotometer (7305 Jenway Staffordshire, UK). In order to calculate the percentage of inhibition, Equation (1) was used. These analyses were performed in triplicate.

Subsequently, the results were expressed as the inhibitory concentration (IC_50_) and the percentage of inhibition of four different concentrations was calculated to obtain a linear regression curve. These analyses were performed in triplicate.
(1)$$\mathrm{Percent}\;\mathrm{inhibition}=\left[\frac{A0-A1}{A0}\right]\;*100$$

A0 = Control absorbance 

A1 = Sample absorbance.

### 2.8. Antioxidant Activity by 2,2′-Azino-Bis-3-Ethylbenzothiazoline-6-Sulfonic Acid Radical Cation (ABTS●+)

The method proposed by Re et al. [26] was used. An ABTS●+ radical cation was formed from 0.0033 g of potassium persulfate and 0.0194 g of ABTS reagent (2,2’-azinobis-(3-ethylbenzothiazolin-6-sulfonic acid (Sigma Aldrich, CAS 30931-67-0, St. Louise, MO, USA) mixed in 5 mL of distilled water. The mixture was stirred and kept in the dark for 16 h at room temperature. After this, a mixture was made with absolute ethanol and the ABTS●+ radical cation until an absorbance of 0.70 ± 0.02 at 754 nm was obtained.

A total of 3920 μL of the ABTS radical solution were then added and the initial absorbance (Ai) was recorded; 80 μL of extract was afterwards added and mixed. After 7 min, the final absorbance (Af) was recorded with the use of a UV-Vis spectrophotometer (7305 Jenway Staffordshire, UK). The inhibition percentage was calculated according to Equation (2) to obtain the medium inhibitory concentration (IC_50_) by a curve of lineal regression using the inhibitory percentage of the four different concentrations. These analyses were performed in triplicate.
(2)$$\mathrm{Percent}\;\mathrm{inhibition}\;\left(\%\right)=\left[\frac{Ai-Af}{Ai}\right]*100$$

### 2.9. Statistic Analysis

The data analysis was performed through a completely randomized design. The experiment consisted of the inoculation of *Pleurotus ostreatus* in two bagasses (two treatments with five replicas) at 25 °C for 28 days. Samples of the substrates were taken at 0, 7, 14, 21 and 28 days of fermentation (5 periods) to obtain the extracts with three organic solvents: acetone and water (80:20 *v/v*), 100% methanol, and water at 100% (3 extracts); the secondary metabolite content and antioxidant activity were determined as well. Samples were analyzed in quadruplicate (*n* = 4).

The results were expressed as a mean and standard deviation, and were examined through the Statistical Analysis System (SAS) version 9.0 statistical package [27]. Additionally, an analysis of variance (ANOVA) means test (Tukey α = 0.05) and Pearson correlation were performed.

## 3. Results

### 3.1. Physicochemical Characterization

Table 1 shows the physicochemical characterization of the residues before fermentation. The values in percent humidity and A*w* are similar since both bagasses were subjected to the same drying process at 60 °C for 12 h and then they were crushed. 

### 3.2. Secondary Metabolites

#### 3.2.1. Quantification of Total Phenolic Compounds (TPC)

In Figure 1, it is shown the total phenolic compounds resulting from the solid-state fermentation of agroindustrial wastes. The TPC had its highest concentration at the beginning of the experiment (day 0) in both residues, and after fermentation it decreased, increasing again after day 21.

Apple bagasse extract (Figure 1a) increased its concentration 21 days after fermentation, with intervals of 0.385 and 0.587 mg gallic acid equivalent/g substrate dw. The highest concentration was present in the aqueous extract with a value of 0.587 ± 0.013 mg gallic acid equivalent/g substrate dw. The TPC content of agave mezcalero bagasse from the acetone and water (80:20 *v/v*) and methanol extracts tended to decrease during fermentation, while the aqueous extract increased its concentration after day 21, until it reached 0.976 ± 0.126 mg gallic acid equivalent/g substrate dw at 28 days.

#### 3.2.2. Total Flavonoid Quantification (TF)

In Figure 2, the total flavonoids in apple bagasse and mezcalero agave are shown. First, the apple increased its TF content from day 21, giving final values (on day 28) of 0.015 ± 0.018, 0.013 ± 0.02, and 0.023 ± 0.012 mg QE/g substrate dw in extracts of acetone and water (80:20 *v/v*), methanol and the aqueous extract, respectively; the latter was the extract with the highest amount of flavonoids. Then, the agave mezcalero bagasse registered a similar tendency to the content of TPC. The total flavonoid concentrations in extracts of acetone and water (80:20 *v/v*) and methanol decreased during fermentation, while in the aqueous extract there was an increase in the TF content at the end of the fermentation time, with a final concentration of 0.012 mg QE/g substrate dw.

#### 3.2.3. Total Triterpenes Quantification (TT)

In Figure 3, the total triterpenes content is shown. Apple bagasse presented TT contents which increased from day 21 in extracts of acetone and water (80:20 *v/v*) and methanol, with final values (on day 28) of 8.043 ± 0.696, and 9.411 ± 2.512 mg ursolic acid/g substrate dw, respectively. The aqueous extract showed steady growth from the beginning of the solid-state fermentation of triterpenes, and reached a concentration of 26.440 ± 0.949 mg ursolic acid/g substrate dw at 28 days. Despite the fact that the TT in agave mezcalero bagasse had the highest value at the beginning of the fermentation, its concentration decreased in all extracts during this process.

### 3.3. Antioxidant Activites

The antioxidant activity of the extracts obtained by the solid-state fermentation of *Pleurotus* from apple and agave mezcalero bagasses was expressed as the 50% inhibitory capacity of the radical in question (IC_50_), which refers to the concentration of antioxidant required to reduce the initial amount of the DPPH and/or ABTS radical by 50%. This is the parameter used to measure the antioxidant properties of a substance [28], that is, the lower the IC_50_ value, the greater its antioxidant activity.

#### 3.3.1. Antioxidant Activity by 1,1-Diphenyl-2-Picril Hydracil (DPPH•)

In apple bagasse, the IC_50_ of the DPPH assay ranged from 14.67 to 385.01 g/L residue. On the other hand, the agave mezcalero bagasse presented IC_50_ values from 61.80 to 377.75 g/L residue (Figure 4a,b). A tendency for IC_50_ to increase was observed from day 0 to day 21, however, a decrease was perceived from day 28 in different extracts.

#### 3.3.2. Antioxidant Activity by 2,2′-Azino-Bis-3-Ethylbenzothiazoline-6-Sulfonic Acid Radical Cation (ABTS●+)

In the ABTS test, the apple bagasse presented a range of residue values from 60.20 to 423.09 g/L; meanwhile, the agave mezcalero bagasse ranged from 74.65 to 390.86 g/L (Figure 5a,b). The obtained results in the apple bagasse showed that the initial values (on day 0) were higher than the agave mezcalero bagasse with intervals of 61.80 to 91.42 g/L of residue.

The acetone and water (80:20 *v/v*) and aqueous extracts increased in IC_50_ during the days of fermentation. It should be noted that the methanol extract displayed a decrease in IC_50_ at day 28, which means that the secondary metabolites produced at the same time of solid fermentation denoted the antioxidant property of this residue.

## 4. Discussion

In both residues, the pH showed values lower than 5.0, lower than the value recommended for the adequate growth of *Pleurotus* (5.5–6.5) [29]. The pH was adjusted for proper fungus growth and fermentation development. The highest amounts of total soluble solids (°Brix), acidity percentage and reducing sugars were found in the apple bagasse because the residue had a significant amount of polysaccharides and organic acids [30]. This limited the growth of *Pleurotus*, thus, bagasse was subjected to a wash with tridestilated water. After washing the apple bagasse released sugars, it was observed that its content of reducing sugars decreased (Table 1), which was resulted in an adequate growth of the *Pleurotus* strain in the residue.

Apple bagasse presented a significant difference in the aqueous extract of the total triterpenes test, having the highest value; the other extracts showed no significant difference in the other tests performed. In the agave mezcalero bagasse, the significant difference between extracts was manifested in the tests of total flavonoids and total triterpenes; the aqueous extract had the highest value, while the others did not show significant differences in the rest of the performed tests.

The highest phenolic content was at the beginning, since the phenolic compounds are shown in the subtracts (bagasses). In addition, it contains lignin, which is a phenolic polymer made up of three principal monomerics: paracoumaryl alcohol, synapilic alcohol and conyferyl alcohol [31,32]. This acts as a carbon source for the *Pleurotus* fungi for its growth and development (trophophase) after the mycelial growth. However, the nutrient exhaustion produces the biosynthesis of secondary metabolites in the idiophase. In this period (T21) there is a metabolite increase according to Ferrer-Romero et al. [33] and Vamanu [34] which supports the presence of *Pleurotus* and its survival in the residue [35]. Therefore, to observe a higher antioxidant activity, idiophase time must be increased in order.

The TPC values generated from *Pleurotous* in other residues, such as the mixture of pineapple and rice straw residues [36], reported values of 0.179–0.650 mg GAE/g substrate dw. Other authors reported greater values than the ones obtained in this research: Hamdipour et al. [37] showed values of 5.53 to 11.6 mg GAE/g dry weight in the fermentation of apple residues with *Rhizopus oligosporus*, and Ajila et al. [38] obtained values of 4.6 to 16.12 mg GAE/g dry weight during the solid-state fermentation of apple residues with *Phanerocheate chrysosporium*. Meanwhile, Ferrer-Romero et al. [33] obtained 0.1 g of total phenols/g of glucose by submerged fermentation from the mycelial biomass of *Pleurotus ostreatus*.

The *Pleurotus ostreatus* strain of this study led to the degradation of lignin to obtain the necessary nutrients for its development, given that growth was observed in both lignocellulosic substrates: apple bagasse and mezcalero agave bagasse [39]. In addition, the genus *Pleurotus* has the capacity to produce laccase enzymes, which have a fundamental role both in the degradation of lignin and in the biosynthesis of phenolic compounds in white-rot fungi [40].

Boonsong et al. [41] conducted an investigation with the purpose of evidencing the high nutritional value of edible fungi. This investigation reported intervals from 2.29 to 2.61 mg QE/g in aqueous extracts of *Pleurotus* strains, which are higher values compared to the ones obtained in this project.

Furthermore, Braga et al. [42] reported significant concentrations of total flavonoids (1.76 mg QE/g) in grape marc residue, and 1.70 mg quercetin equivalent/g in mango residue; Dulf et al. [43] found slightly lower values of 0.29 and 0.36 mg QE/g, resulting from the solid-state fermentation of apricot residue with *Aspergillus niger* and *Rhizopus oligosporus* for 14 days; while Lessa et al. [44] found concentrations of TF with values of 31.5 mg QE/100 g in fermented cocoa flour by means of *Penicillium roqueforti*. Therefore, the low concentration of this type of metabolites maybe be determined, once again, by the used strain, that degrades lignin during solid-state fermentation.

Yang et al. [45] found a higher concentration of triterpenes, 47.10 mg/g, 30 days after the solid-state fermentation of citrus residues with the fungus *Antrodia cinnamomea*. In this investigation, the total triterpenes values have great importance, especially those found during solid fermentation in the apple bagasse because this type of secondary metabolite has antibacterial properties against plant pathogens; in addition, it has demonstrated effectiveness against oxidative damage through the elimination of free radicals and the modulation of enzymatic activity [46]. The authors mention solvents with low polarity for the extraction of triterpenes, such as ethyl acetate [47,48]. Nevertheless, aqueous extracts have been used to identify secondary metabolites, like triterpenes, in other macromycete fungi, *Phellinus* and *Ganoderma* [49]. As a consequence, it was decided to use this type of solvent, due to polyphenols, sterols, triterpenes and anthracenes being moderately abundant [50].

The production of secondary metabolites can be considered similar to the tendency of antioxidant activity, due to the fact that since day 21 there was an increase in antioxidant activity in the apple bagasse, which can be attributed to a higher concentration of polyphenols present in acetone and water (80:20 *v/v*) and aqueous extracts. As mentioned by Lu and Foo [51], the antioxidant activity is caused by polyphenols in apple residues. Meanwhile, Ajila et al. [38] report IC_50_ values of 12.24 ± 5.20 to 14.27 ± 2.50 µg in acetone and water (80:20 *v/v*) extracts and 40.16 ± 2.40 to 50.18 ± 2.00 µg in aqueous extracts in fermented apple residue.

While there is little information on antioxidant activity in agave mezcalero through solid-state fermentation, some authors have found this property in certain agave species. For example, Carmona et al. [52] mention the antioxidant activities of *Agave lechuguilla*, and Araldi et al. [53] mention it regarding *Agave sisalana*. Both species are used as a source of raw material in the production of biofuels from solid fermentation.

Although the values obtained can be considered low, the obtained methanol extract from the solid-state fermentation of the agave mezcalero can be used as an alternative for the production of compounds with antioxidant properties.

Overall, the IC_50_ values of apple bagasse are lower; they displayed greater inhibitory activity against the DPPH radicals and ABTS radicals in comparison to those of the agave mezcalero bagasse. On the other hand, Ignat et al. [54] express that the different results in the metabolite content may be due to the extraction conditions, type of solvent, and its susceptibility to degradation. This way, it is possible to indicate the variability of secondary metabolites and antioxidant activity in agroindustrial waste.

## 5. Conclusions

The growth of *Pleurotus ostreatus* was observed in both the apple bagasse and agave mezcalero bagasse residues due to their ability to secrete extracellular, non-specific ligninolytic enzymes during secondary metabolism. Spectrophotometric studies have demonstrated that, because of solid-state fermentation, there was presence of secondary metabolites—phenolic compounds, flavonoids and triterpenes.

The highest concentration of compounds of biological interest occurred in apple bagasse after 21 days, with the aqueous extract presenting the highest extraction of the different phytochemical compounds. This behavior was reflected in the antioxidant activity of the different extracts during the same fermentation time. Minor amounts of phenolic and flavonoid compounds were found in the aqueous extract of the agave bagasse from day 21 of fermentation.

The biosynthesis of secondary metabolites displaying antioxidant activities in the solid-state fermentation of *Pleurotus ostreatus* in the apple residues of the cider industry and in the agave residues of the mezcalera industry can contribute to the use of these wastes, as they can be used in functional food formulation or, due to their medical properties, in the pharmaceutical industry.

## Figures and Tables

**Figure 1 jof-06-00137-f001:**
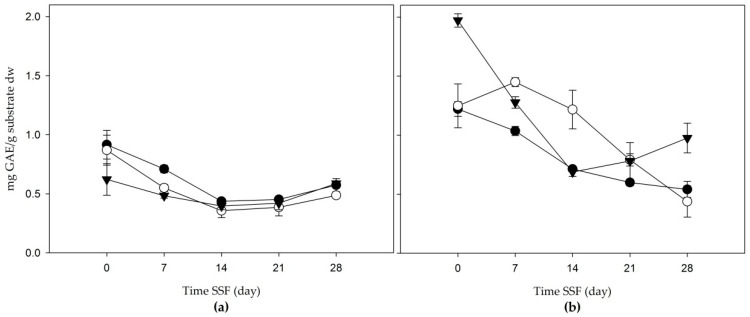
Total phenolic compounds (mg gallic acid equivalent per g substrate on dry weight (mg GAE/g substrate dw)) of the solid-state fermentation of agroindustrial residues: (**a**) Apple bagasse; (**b**) Agave mezcalero bagasse. Acetone and water (80:20 *v/v* (
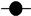
)); methanol extract (
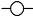
); aqueous extract (
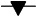
). Time SSF = time solid-state fermentation. Values are the mean ± standard deviation.

**Figure 2 jof-06-00137-f002:**
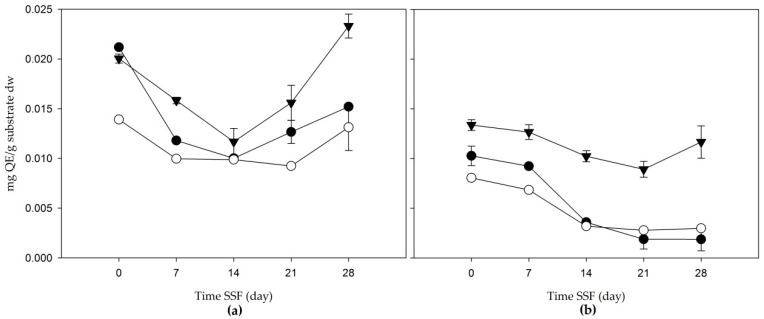
Total flavonoids (mg QE/g substrate dw) of the solid-state fermentation of agroindustrial residues: (**a**) apple bagasse; (**b**) agave mezcalero bagasse. Acetone and water (80:20 *v/v* (
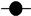
)); methanol extract (
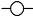
); aqueous extract (
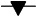
). Time SSF = time solid-state fermentation. Values are the mean ± standard deviation.

**Figure 3 jof-06-00137-f003:**
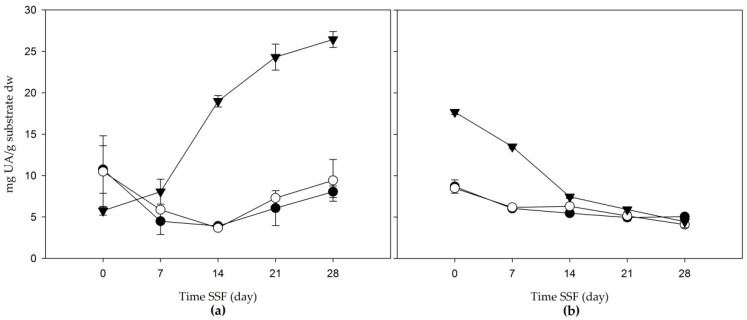
Total triterpenes (mg UA/g substrate dw) of the solid-state fermentation of agroindustrial residues: (**a**) apple bagasse; (**b**) agave mezcalero bagasse. Acetone and water (80:20 *v/v* (
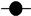
)); methanol extract (
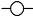
); aqueous extract (
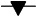
). Time SSF = time solid-state fermentation. Values are the mean ± standard deviation.

**Figure 4 jof-06-00137-f004:**
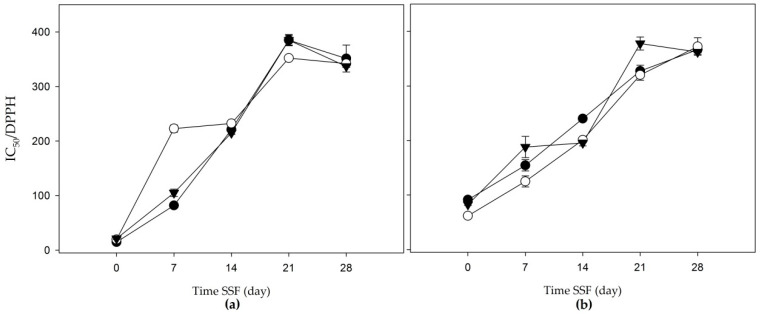
50% inhibitory concentration in 2,2-diphenyl-1-picrylhydrazyl (DPPH (IC_50_/DPPH)) of the solid-state fermentation of agroindustrial residues: (**a**) apple bagasse; (**b**) agave mezcalero bagasse. Acetone and water (80:20 *v/v* (
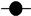
)); methanol extract (
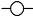
); aqueous extract (
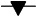
). Time SSF = time solid-state fermentation. Values are the mean ± standard deviation.

**Figure 5 jof-06-00137-f005:**
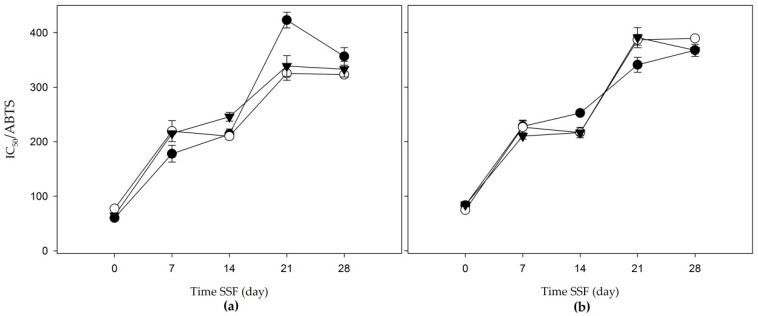
A 50% inhibitory concentration in 2, 2′-azino-bis (3-ethylbenzothiazoline)-6 ammonium sulfonate (ABTS (IC_50_/ABTS)) of the solid-state fermentation of agroindustrial residues: (**a**) apple bagasse; (**b**) agave mezcalero bagasse. Acetone and water (80:20 *v/v* (
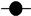
)); methanol extract (
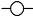
); aqueous extract (
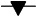
). Time SSF = time solid-state fermentation. Values are the mean ± standard deviation.

**Table 1 jof-06-00137-t001:** Physicochemical characteristics of apple bagasse and agave mezcalero bagasse.

Agroindustrial Residue	°Brix	pH	Moisture Percentage (%)	A*w*	Titratable Acid (%)	Direct Reducing Sugars ^1^ (%)	Direct Reducing Sugars ^2^ (%)
Apple bagasse	8.90 ± 0.28 ^a^	4.04 ± 0.15 ^b^	8.67 ± 0.20 ^a^	0.98 ± 0.05 ^a^	0.97 ± 0.23 ^a^	86.20 ± 7.11 ^a^	9.98 ± 0.01 ^a^
Agave mezcalero bagasse	0.36 ± 0.20 ^b^	5.00 ± 0.10 ^a^	8.68 ± 0.20 ^a^	0.97 ± 0.01 ^a^	0.87 ± 0.21 ^a^	6.79 ± 0.37 ^b^	6.79 ± 0.37 ^b^

Values are the mean ± standard deviation. Means with the same letters within each column do not differ statistically by Tukey’s test (*p* ≤ 0.05). ^1^ Direct reducing sugars before washing apple bagasse; ^2^ direct reducing sugars after washing apple bagasse.

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
