# Peer review of "Secondary Metabolites and Antioxidant Activity of the Solid-State Fermentation in Apple (Pirus malus L.) and Agave Mezcalero (Agave angustifolia H.) Bagasse"

_jof, 2020, doi:10.3390/jof6030137_

Round 1

Reviewer 1 Report

The manuscript “ Secondary metabolites and antioxidant activity of the solid-state fermentation in apple (Pirus malus L.) and agave mezcalero (Agave angustifolia H.) bagasse ” should be an interesting study to demonstrate the potential of solid state fermentation to produce antioxidant compounds. But the manuscript is not acceptable for publication as major modifications and corrections have to be done, as detailed below.

The authors have to highlight the novelties of this study. Is it the first used of these two substrates by SSF? Is Pleurotus ostreatus well known to produce metabolites ? Why triterpenes ? Such metabolites are known to be  produced by the fungus ? Which kind of triterpenes ? Some recent and relevant References must be included and commented in the Introduction.

Line42 : It should be  2,2-Diphenyl-1-picrylhydrazyl

Line43 : It should be  2,2′-azino-bis-3-ethylbenzothiazoline-6-sulphonic acid radical cation (ABTS●+) assay

« Physicochemical tests were performed to the crushed bagasse in order to know its attributes and adjust its physicochemical properties to make them suitable for the growth of the Pleurotus fungus » What are these suitable conditions ? aw, humidity, temperature ? Were temperature and moisture content controlled during the process ?

Line92 : 80% acetone (v/v) : and 20% of water ?

Line92 : MeOH is toxic, why do you not used EtOH ?

Line128 : The DPPH solution in EtOH ?

How the IC50 of the extracts were calculated from the inhibition percentages ?

For DPPH test, BHA, BHT or  ascorbic acid were generally used as positive controls. Which one do you used?  This is important to ensure that the test has been successfully completed.

Line 141and 145 : It should be   ABTS●+radical cation

Line 150 : How the IC50 of the extracts were calculated from the inhibition percentages ?

Line 174 : Since the higher content in total phenolic content is observed at T0 what is the interest of the fermentation? If phenolic compounds are degradation products of the substrate, why the concentration decrease? What is the explanation?

Figure 3 : Is it normal to have more triterpenic compound in the aqueous extract? How can you justify this?

Line 250 : AOA depends not only on the concentration of active metabolites but also on the relative proportion of each of the constituents. As you don’t have the composition of the various mixture  and since Folin Ciocalteu is not specific of the phenolic compounds it’s impossible to conclude. So you should delate Table 2.

P281 : lycopene is not a phenolic compound!

And more generally authors do not present a discussion but a list of articles with results that have no real connection with the study (other substrates and other fungus).

Reviewer 2 Report

The paper entitled "Secondary metabolites and antioxidant activity of the
solid-state fermentation in apple (Pirus malus L.) and agave mezcalero (Agave angustifolia H.) bagasse" deals with intersting topic - usage of waste obtained from food industry in solid-state fermentation process. 

However, the paper lacks some important results from my point of view and also, there some problematic statements provided:

  1. Authors stated that the humidity of substrates used in SSF was adjusted to 80% (line 84), but in table 1 there is percentage of humidity only 8,6. I am not sure the table 1 concerns the subtrate after sterilization or just plain substrate
  2. line 280 - authors stated that "Assi and King found quantities of micrograms per gram of phenolic compounds, such as α-tocopherol,
    lycopene and β-carotene in tomato pomace..." None of these compound are phenolic compounds - tocopherol is vitamin E and lycopene and betacarotene are carotenoid pigments, so it is not possible to compare the content of such compounds with phenolic compounds 
  3. I would prefer if authors provided the data from DPPH and ABTS measurements expressed as equivalent of trolox, since it would be much more clear and also it would be easier to compare the results from data of other authors. Also, in this case, the authors would avoid to the negative correlation results between antioxidant activity and content of phenolics and other compounds.
  4. The paper does not stated or described the rate of biomass growth. The statement in line 332 "Adequate growth of Pleurotus ostreatus was observed in both the apple bagasse and agave" is absolutely not sufficient. In my opinion the rate of biomass growth in SSF is directly connected to all parameters that authors provided in results.
  5. All the results of measurement of phenolics, flavonoids and triterpenes are only from spectrophotometry of different extracts. On one hand, these methods are very well established and used, but on the other hand, authors should considers that there also other compounds in extracts that can completely interfere with there results - especially in correlation with antioxidant capacity measurements. The water extracts definitely contained also the proteins, which has been described many times before as good antioxidants. Could authors provided at least some data of composition of extracts - obtained for example at least by simple HPLC measurement? So it would be clear that there are specific compounds in the SSF products?

Reviewer 3 Report

General comments: The study reported by the authors is a god study. However, there is a great requirement for sentence and grammar corrections of this manuscript. The inconsistencies and grammatical errors make this manuscript decrease the readability. Additionally, the context is slightly vague (as can be seen in abstract, Lines 24-26). This aspect needs to be highlighted properly. This has been mentioned to a degree introduction (Lines 53-62), but is missing from abstract. Due to this, the study loses its 'so what?' perspective (as in: authors have reported the yield of x quantities of antioxidants from biomass. However, what's the use of these techniques towards agriculture, water treatment etc? Can they be used as pre-treatment methods? Can they be used to extract commercial compounds? anything else?). This needs to be presented well in the abstract and discussion part.

Overall, this manuscript needs a major correction, especially in grammatical area before it can be accepted for publication.

Specific comments:

Line 69: Is the term 'scalded' correctly used here? please review. Also, when indicating a product (air dryer here), the product details should be mentioned (Model number, Manufacturer, city, Country)

Line 70: Replace 'humidity' with 'moisture content'

Line 94 and elsewhere: The spinning centrifugation must always be stated in 'g' or 'rcf values'. RPM is an incorrect way to indicate this. Please correct

Line 100: Is the term (m/v) correct? Are you referring to (w/v) (weight/volume)?

Line 103 and elsewhere: mg/mL must be replace by g/L or similar SI equivalent units

Line 222: The term 'antioxidant power' is incorrect. Do you mean 'antioxidant activity'?

Round 2

Reviewer 1 Report

The autors  have taken into account my remarks and have answered to the different points. However there are still little mistakes :

-the médium inhibitory : medium

-In adittion : in addition

-polymere : polymer

-synapilic alcohol  : sinapilic

-tripertens : triterpens?

-that is, the lower the IC50 value, the greater its antioxidant activity : to be replace by A smaller IC50 means higher antioxidant activity.

Reviewer 2 Report

The manuscript has been improved, however, there is lack of discussion concerning the changes in triterpens.

There is still lack of discussion that would concern the presence of other compounds in extracts - such as proteins in water extracts. I do not believe that there were no proteins presented, since authors clearly stated that their fungus produces extracelullar enzymes that cleave lignin.

Also, authors have stated in conclusion that such agricultural wastes can be used as source of interesting compounds, but according to their data, the non-fermented waste has the highest of all analysed compounds (phenolics, flavonoids, triterpens).